# Interplay of Vitamin D, Unfolded Protein Response, and Iron Metabolism in Neuroblastoma Cells: A Therapeutic Approach in Neurodegenerative Conditions

**DOI:** 10.3390/ijms242316883

**Published:** 2023-11-28

**Authors:** Gergely Jánosa, Edina Pandur, Ramóna Pap, Adrienn Horváth, Katalin Sipos

**Affiliations:** Department of Pharmaceutical Biology, Faculty of Pharmacy, University of Pécs, H-7624 Pécs, Hungary; janosa.gergely@gytk.pte.hu (G.J.); pap.ramona@pte.hu (R.P.); horvath.adrienn2@pte.hu (A.H.); katalin.sipos@aok.pte.hu (K.S.)

**Keywords:** vitamin D, unfolded protein response, UPR, iron, SH-SY5Y, cell culture

## Abstract

Vitamin D3 (VD) is crucial for various cell functions, including gene regulation, antioxidant defense, and neural health. Neurodegenerative conditions are closely linked to the unfolded protein response (UPR), a mechanism reacting to endoplasmic reticulum (ER) stress. Iron metabolism is intricately associated with UPR and neurodegeneration. This study used SH-SY5Y neuroblastoma cells to investigate the relationship between UPR, iron metabolism, and VD. Different sequences of treatments (pre- and post-treatments) were applied using VD and thapsigargin (Tg), and various methods were used for evaluation, including real-time qPCR, Western blotting, ELISA, and iron content analysis. The findings indicate that VD affects UPR pathways, cytokine release, and iron-related genes, potentially offering anti-inflammatory benefits. It also influences iron transporters and storage proteins, helping to maintain cellular iron balance. Furthermore, pro-inflammatory cytokines like interleukin-6 (IL-6) and tumor necrosis factor alpha (TNFα) were impacting UPR activation in cells. VD also influenced fractalkine (*CX3CL1*) gene expression and secretion, suggesting its potential as a therapeutic agent for addressing neuroinflammation and iron dysregulation. This research provides insights into the intricate connections among VD, UPR, and iron metabolism in SH-SY5Y neuroblastoma cells, with implications for future investigations and potential therapeutic approaches in neurodegenerative diseases characterized by UPR dysregulation and iron accumulation.

## 1. Introduction

Vitamin D3 (VD), also known as cholecalciferol, is an essential micronutrient that has a key function in cellular metabolism. It is crucial for managing calcium balance and supporting bone well-being [1,2]. Recent insights suggest it influences several cellular activities, such as cell proliferation, differentiation, and programmed cell death [1,3]. Research indicates that VD modulates gene activity, impacting around 200 genes across different tissues [4]. The active variant of VD, 1,25-dihydroxy vitamin D [1,25(OH)2D], interacts with the vitamin D receptor (VDR) found in numerous cells and tissues. This interaction between 1,25(OH)2D and VDR affects gene expression, resulting in various biological outcomes. Beyond gene regulation, VD contributes significantly to cellular vitality, protecting against abnormalities. It offers antioxidant and anti-inflammatory benefits, decreasing oxidative stress in cells and consequential and dissuasive damage [5]. VD is invaluable for neural health, supporting brain development, shielding from neurodegenerative conditions, and enhancing cognitive performance [6]. Moreover, VD influences gene activity linked to neurotransmitter production, synaptic plasticity, and maintenance of neuronal structures [7].

The unfolded protein response (UPR) is a universally acknowledged cell mechanism regulating metabolic activities. It becomes active by the presence and accumulation of improperly folded proteins in the endoplasmic reticulum (ER), which is the cellular component overseeing protein folding and quality [8,9]. Disturbances in ER stability can generate ER stress, which can arise from various factors like inadequate nutrients, infections, oxidative distress, calcium discrepancies, and more. Once these stress indicators are perceived, UPR mechanisms mobilize to restore ER balance and foster cell resilience. The primary UPR pathways are steered by proteins like IRE1, PERK, and ATF6, each managing specific downstream actions to enhance protein arrangement and response to ER stress [10]. Exposure to certain substances can promote UPR induction. For instance, thapsigargin (Tg), which affects ER calcium levels, is commonly used in studies related to diseases like Alzheimer’s. Tg exerts its molecular effect by inhibiting the sarcoplasmic/endoplasmic reticulum calcium ATPase (SERCA), causing ER calcium deficits and activation of UPR [11].

Moreover, the UPR oversees other cellular functions like cell longevity, autophagy, and inflammatory response [12]. It aids in cell survival by upregulating specific genes and curbing oxidative stress. UPR also triggers the degradation of damaged organelles, ensuring cellular stability [13]. Aberrations in the UPR can be linked to diseases like Alzheimer’s and Parkinson’s [8,14]. Hence, understanding UPR is vital for developing potential treatments.

Iron is indispensable for the neural system, aiding neurotransmitter production, myelin creation, and brain metabolic processes [15,16]. Discrepancies in iron balance are associated with neurodegenerative diseases, leading to excess iron accumulation in the brain, which catalyzes oxidative stress and contributes to neuronal damage. This imbalance in iron homeostasis, linked to protein aggregation and neuroinflammation, exacerbates the progression of these diseases; therefore, achieving an optimal iron balance is pivotal [15,17,18]. Furthermore, cytokines significantly influence general health by managing immune reactions, inflammation, and more, and iron is integral to this complex interplay [19].

Inflammatory cytokines like IL-1, IL-6, and TNF-α affect iron balance through the elevation of hepcidin, leading to altered iron absorption. This mechanism restricts iron during inflammation, preventing potential harm. The link between inflammatory cytokines and UPR is labyrinthine, with cytokines potentially inducing UPR pathways, thereby contributing to proper protein folding and ER stability during inflammatory conditions [20,21].

Fractalkine, also referred to as *CX3CL1*, is a unique chemokine vital for immune cell recruitment and activation by binding to the *CX3CR1* receptor. It is fundamental in processes like inflammation, immune cell migration, and neuronal responses. Its role in the nervous system is paramount, affecting brain development, synaptic adaptability, and reactions to neurodegenerative diseases [22]. Importantly, interactions have been observed between iron, VD, and the regulation of fractalkine, emphasizing the interplay between VD, iron, and neuronal immune functions [23,24,25]. VD supplementation has been shown to decrease fractalkine levels in airway smooth muscle cells, particularly when administered in conjunction with fluticasone, a widely used asthma medication [23]. In immune cells like monocytes, fractalkine modulates iron levels and inflammatory responses, particularly under conditions that elicit an immune response [24]. In the brain, fractalkine affects microglial cells, leading to increased iron accumulation in neurons, a factor that might be implicated in the pathogenesis of neurodegenerative diseases [25]. The interrelationship between VD, UPR, and iron in the neural system remains perplexing and not fully understood. Alongside the well-known VDR receptor of VD, there is a newly acknowledged membrane receptor, protein disulfide-isomerase A3 (*PDIA3*). This receptor might account for the immediate actions of VD [26]. Furthermore, while VD modulates genes related to iron equilibrium and UPR, the UPR’s activation in response to cellular anomalies may affect iron’s accessibility for neuronal processes. Comprehensive research is imperative to elucidate the interplay between VD, UPR, and iron dynamics within the nervous system.

We aimed to explore UPR’s influence on iron balance and understand how pre- and post-administration VD might mitigate the stress caused by UPR concerning iron homeostasis.

## 2. Results

### 2.1. Verification of UPR Generated in SH-SY5Y Cell by Tg Administration

In order to verify the development of UPR, we examined the gene expression of UPR-related genes, such as *DDIT3*, *HSPA5*, and *XBP1*. After the Tg treatment, the relative expression of every investigated gene increased significantly compared to the control (Figure 1A). To avoid any miscellaneous cause of this elevation, we also analyzed the same elements in the VD-treated samples. In this case, no such increase was detected (Figure 1A). The analysis of protein expression showed similar results. The addition of Tg multiplied the amount of the target proteins (CHOP, BiP, and XBP-1s), while no significant changes were detected in the VD-treated group (Figure 1B).

### 2.2. Expression Analysis of PDIA3 in SH-SY5Y Cells after Tg Treatment

*PDIA3*, also termed ERp57, belongs to the protein disulfide-isomerase (PDI) family and is pivotal in the facilitation of protein structuring and surveillance within the ER. Consequently, any dysregulation of *PDIA3* could be linked to the initiation of the UPR. In the Tg-administered cells, the *PDIA3* gene’s relative expression has increased significantly (Figure 2A). On the contrary, the level of Erp57 protein encoded by the *PDIA3* gene did not change compared to the untreated control group (Figure 2B).

### 2.3. Effects of VD or Tg on the mRNA Expression of the HAMP Gene

The protein Erp57 has many cellular roles, such as acting as a cofactor of iron metabolism pathways. Changes in these signal transduction pathways can affect the expression of the *HAMP* gene, which encodes the key hormone regulating iron homeostasis, hepcidin. Alterations in this gene upon Tg and VD treatment are already known [27,28,29]. In contemplation of obtaining the same result, we examined the *HAMP* gene on the mRNA level. After Tg administration, we experienced skyrocketing growth, while VD treatment alone resulted in a lower expression compared to the control (Figure 3). These results are in agreement with the published ones.

### 2.4. Administrations of VD and Tg in Different Orders Produced Alterations in the Expression Levels of UPR-Related Genes

Our research intrigued us to see if the treating agents are administered after each other in a different order. We analyzed the UPR-related genes in the context of pre- and post-treatment of VD and Tg. Although pre-treatment with VD did not abolish the effect of Tg, it resulted in a reduction in the expression of the target genes, with a significant difference for *DDIT3* compared to the Tg treatment alone (Figure 4A). On the other hand, VD post-treatment diminished the relative expression for each gene to the level of control (Figure 4A). Compared to the mRNA expression, similar results were observed for Tg treatment at the protein level, where we also saw a significant increase in the expression of each gene for each protein examined (Figure 4B). Unlike the mRNA levels of the VD post-treated cells (Tg+VD), among the proteins, only XBP1s showed a substantial decrease compared to the Tg-treated cells (Figure 4B).

### 2.5. VD Post-Treatment Resulted in Reduced Expression of PDIA3

Since *PDIA3* is an important receptor of VD, we have also determined its mRNA expression for co-treatments. In the VD+Tg group, we experienced the same high gene expression as in the samples treated with Tg only (Figure 5A). Contrariwise, the VD post-treatment resulted in decreased mRNA levels compared to the Tg group (Figure 5A). However, the protein expression showed a completely different pattern. In both VD pre- and post-treatment, Erp57 protein expression was significantly reduced compared either to the control or to the Tg group (Figure 5B).

### 2.6. Post-Treatment with VD Suppressed the Expression of the HAMP Gene

We examined the effects of the combinatory treatments on the mRNA expression levels of the *HAMP* gene and could see a similar result as in the case of the *PDIA3* gene. The relative expression in the VD pre-treated group was increased like in the aforementioned Tg treatment, but the administration of VD after Tg treatment yielded an mRNA level equivalent to the control (Figure 6).

### 2.7. Expression Changes in Iron Transporters, Ferroportin, and DMT1 in the Different Treatments of SH-SY5

As the expression level changes in the *HAMP* gene prove, the order of the treating agents can influence the key regulator of iron metabolism; therefore, we looked at the behavior of iron transporters. We examined the mRNA and the protein level of the DMT1, which is responsible for iron uptake, and the FPN, which is involved in the iron export of the cell. In the context of gene expression, neither pre- nor post-treatment with VD made a significant difference (Figure 7A). The relative expression of DMT1 was statistically reduced compared to Tg administration only (Figure 7A). In the protein analysis, a decrease was observed in the VD pre-treated sample in both FPN and DMT1 (Figure 7B).

### 2.8. Iron Storage and Cellular Iron Content after VD and Tg Administration

To investigate the possible changes in the cytosolic iron store, analyses of the FTH were carried out at mRNA and protein levels. The Tg and VD+Tg groups displayed elevation in the relative expression of FTH compared to the control, while the VD-only treatment and VD post-treatment resulted in low mRNA expression in comparison with the Tg treatment (Figure 8A). At the protein level, a significant decrease was detected after Tg administration and at VD post-treatment (Figure 8B). Nevertheless, there was a statistically verifiable increase in the VD-only and VD-pre-treated groups compared to the Tg administration alone (Figure 8B). In addition, we determined the total iron content of the cells. Although we did not obtain the same statistically verifiable differences as at the FTH levels, a similar pattern was observed (Figure 8C).

### 2.9. Changes in the Secreted Proinflammatory Cytokine Levels of the Treated SH-SY5Y Cells

Cytokines play a regulatory role in both iron metabolism and the UPR, particularly in the context of neurodegenerative disorders. Proteins like IL-6 and TNFα are known to alter hepcidin levels and exert an effect on neuronal UPR, outcomes of which may range from protective to harmful based on the extent of UPR activation. We determined the concentration of the secreted IL-6, interleukin-8 (IL-8) and TNFα. The results showed a pattern in all three cases. In the Tg group, elevated cytokine concentrations were measured for all three cytokines (Figure 9A–C). For IL-6 and IL-8, pre- and post-administration of VD significantly lowered the secretion of these proinflammatory cytokines (Figure 9A,C).

### 2.10. Expression of mRNA and Protein of Fractalkine after the Tg and VD Treatments

Fractalkine has been shown to regulate iron metabolism and has also been implicated in the UPR as a chemoattractant signal molecule. In the context of gene expression, VD post-treatment resulted in nearly the same outcome as the previously examined genes, like *PDIA3*. Relative expression of *CX3CL1* was increased in the groups with Tg treatment alone and VD pre-treatment (Figure 10A). In the post-dosage of VD, a significant reduction in gene expression was observed compared to the Tg-only treatment (Figure 10A). Interestingly, VD by itself reduced the fractalkine expression. The protein analysis revealed a statistically verifiable increase in VD pre-treatment compared to the Tg administration alone (Figure 10B). Again, it was unusual that VD treatment alone increased the fractalkine protein secretion.

### 2.11. Phosphorylation of Significant Signaling Pathways Components after UPR Generation and VD Effect

The relationship between the unfolded protein response (UPR), VD, and iron metabolism involves intricate crosstalk among signaling pathways. UPR activation can influence the NF-κB, which is a transcription factor complex consisting of p105, a propeptide of p50 and p65 subunits, which regulates the expression of genes involved in immune responses, inflammation, and cell survival. The NF-κB pathway, such as the STAT3 pathway, can be activated by pro-inflammatory cytokines, and it has a crucial role in iron metabolism by regulating the expression of hepcidin, iron transporters, and iron storage proteins, thereby influencing iron absorption, distribution, and utilization in response to inflammatory stimuli. VD, through its interaction with the NF-κB pathway, STAT3 pathway, and UPR by the Erp57 protein, can modulate iron metabolism by regulating hepcidin expression, thereby impacting iron homeostasis. In the case of activated (phosphorylated) STAT3 (pSTAT3), a significant reduction was measured in only the VD-treated cells compared to the Tg administration alone. However, the relative expressions of p105 and p65 were increased in this group. Among the co-treatments, changes were only observed in the VD pre-treated group, with positive differences for p105 and negative differences for p65 compared to Tg-treated cells (Figure 11).

## 3. Discussion

Neurodegenerative diseases, increasingly recognized as significant social and medical burdens, are driven by complex mechanisms often involving the accumulation of proteins and iron, though their precise initiations are not always known [30,31]. This backdrop set the stage for investigating the relationship between the UPR and iron metabolism in neurons. Our research uncovered a multifaceted interplay involving UPR, iron metabolism, and vitamin D, revealing that vitamin D critically modulates UPR signaling pathways in neurons. This modulation, along with vitamin D’s potential anti-inflammatory effects, influences both gene and protein expression related to iron metabolism and UPR. The findings highlight the intricate nature of these interactions and suggest that the timing of vitamin D administration could be key in developing therapeutic approaches for neurodegenerative diseases marked by UPR dysregulation and iron accumulation.

In order to carry out our investigations, we utilized SH-SY5Y human neuroblastoma cells, a well-established model in neurological research [32,33,34]. Though protein accumulation in these cells is studied [35,36], the experimental results of the generation of UPR are limited. In vitro, UPR can be produced in many ways, like oxidative stress, inhibition of post-translational modifications of proteins, inflammation, or changes in the calcium-ion flux of the endoplasmic reticulum. The latter method is especially interesting in the case of neurons, as Ca2+ is involved in action potential and neurotransmitter release [37]. Our objective was to observe and examine the relationship between UPR induced by thapsigargin, vitamin D, and iron homeostasis (Figure 12).

VD, or cholecalciferol, is an essential micronutrient involved in various cellular processes. It regulates gene expression of over 200 genes and has been shown to be important for cell growth, differentiation, and apoptosis [38,39,40]. Additionally, it provides antioxidant and anti-inflammatory effects, protecting against cellular oxidative stress and damage. The role of VD in maintaining the health of the nervous system is particularly noteworthy, as it supports brain development [41], protects against neurodegenerative diseases [42], and regulates the expression of genes involved in neurotransmitter synthesis [43].

VD has been shown to modulate UPR signaling pathways, influencing protein folding and quality control in the central nervous system [44]. In the scientific literature, there is evidence that underscores the potential utility of vitamin D in the management of diseases such as multiple sclerosis (MS), neuromyelitis optica spectrum disorder (NMOSD), Parkinson’s disease (PD), and Alzheimer’s disease (AD). It emphasizes the significance of supplementing vitamin D in individuals to attain levels that may confer therapeutic advantages [45,46]. Other studies demonstrate that a 12-month regimen of daily 800 IU vitamin D supplementation can significantly improve cognitive function and decrease amyloid beta-related biomarkers in Alzheimer’s disease patients, suggesting a beneficial role of vitamin D in managing AD symptoms and advocating for larger, long-term trials to further explore these effects [47]. Therefore, investigating the effects of pre- and post-treatment with VD on UPR activation and subsequent protein handling in the brain could provide valuable insights into potential therapeutic strategies for neurodegenerative diseases associated with UPR dysregulation. The scientific literature contains a wealth of studies investigating the potential benefits of VD pre-supplementation in the context of neurodegenerative diseases, including its effects on UPR and brain health. In a mouse model of Alzheimer’s disease, vitamin D supplementation was found to enhance neurogenesis and cognitive function, with its effectiveness varying by disease stage and gender [48]. In a Parkinson’s disease model, vitamin D effectively protected brain mitochondria from dysfunction and oxidative damage, highlighting its potential as a therapeutic agent for neurodegenerative diseases [49]. However, there is a notable scarcity of research focused on post-treatment interventions in human neuronal aspects with VD at the cellular level, and the limited existing evidence suggests that such approaches have not yielded significant therapeutic outcomes in relation to UPR modulation or related neurodegenerative conditions.

ERp57/PDIA3, a protein, has been detected in the nervous system, and multiple studies have demonstrated its association with neurodegenerative processes [50]. Its association with neurodegeneration has been specifically attributed to its involvement in ER stress pathways, which are known to be associated with diseases like Alzheimer’s disease (AD) [51,52]. Additionally, a hypothesis suggests that the role of ERp57/PDIA3 in neurodegenerative diseases may be linked to its function as a receptor for VD [53]. The results suggest that *PDIA3*, being an important receptor of VD, exhibits distinct patterns of mRNA expression based on different treatments. In the VD+Tg group, the mRNA expression levels remained high, similar to the samples treated with Tg alone. However, in the case of VD post-treatment, there was a decrease in mRNA levels compared to the Tg group. Interestingly, the protein expression of ERp57 displayed a contrasting pattern. Both VD pre- and post-treatment led to a significant reduction in ERp57 protein expression compared to both the control and Tg groups (Figure 5). The results obtained indicate that the downregulation of Erp57 suppresses STAT3 phosphorylation, which can be seen in the only VD-treated group. However, the administration of Tg increases STAT3 phosphorylation, so presumably, to compensate for this, a further decrease in Erp57 levels is observed in the co-treatments (Figure 11).

The intricate interplay between ERp57 and iron metabolism has been established via its interactions with the STAT3 and NFκB signaling pathways [54,55,56,57]. We embarked on an investigation into the potential alterations in iron homeostasis within specific neurodegenerative diseases. By closely examining the dynamics of iron metabolism, we aimed to unravel any potential changes that might be associated with the pathogenesis and progression of these complex neurological disorders. In this study, we investigated the effects of combinatory treatments on the mRNA expression levels of the *HAMP* gene, which plays a crucial role as a key regulator of iron metabolism. The results demonstrated that the order of treating agents can influence the expression of the *HAMP* gene. Pre-treatment with VD (VD+Tg) increased the relative expression of *HAMP* gene mRNA, resembling the effect observed with Tg treatment alone. However, when VD was administered after Tg treatment, the mRNA expression level of the *HAMP* gene returned to levels equivalent to the control group. These results suggest that VD post-treatment resolves the iron retention induced by UPR at the mRNA level.

Intracellularly, iron distribution is meticulously managed across various organelles, each playing a unique role in iron homeostasis. Mitochondria are key organelles that utilize iron for essential roles, notably in synthesizing components like heme and iron-sulfur clusters, which are critical for ATP production [58]. The cytosol plays its part by using proteins like FTH to control the storage and dispersion of iron, thus striking a balance to avoid excess or deficiency. Additionally, organelles such as the endoplasmic reticulum and the Golgi apparatus take part in producing and distributing proteins rich in iron. Iron is also indispensable for the nucleus, especially for vital activities such as the synthesis and repair of DNA [59]. This sophisticated, compartmentalized regulation of iron, involving a network of transporters and regulatory proteins, is crucial for cellular metabolism, highlighting the intricate nature of iron homeostasis within cells. The interconnection of these mechanisms with neurodegenerative conditions and treatment strategies, such as those involving the *HAMP* gene and vitamin D, underscores the complexity and significance of iron regulation in health and disease.

Given the influence of the treatment order on the key regulator of iron metabolism, we investigated the behavior of iron transporters, specifically DMT1 (responsible for iron uptake) and FPN (involved in iron export from the cell). Analysis of gene expression revealed that in the relative expression of DMT1, a statistically significant reduction was observed in the VD post-treated group compared to Tg administration alone but not in the expression of FPN (Figure 7A). All this assumes cells attempt to restore normal metabolism of iron uptake and iron export after a stress state has occurred. Protein analysis also showed interesting results. In the VD pre-treated group, a decrease in both FPN and DMT1 protein expression was observed (Figure 7B). This indicates that VD pre- and post-treatment may impact the levels of these iron transporters at the protein level, potentially affecting iron homeostasis within the cell.

To further investigate potential changes in iron utilization, we examined the expression of FTH, a protein involved in cytosolic iron storage, at both the mRNA and protein levels. The results revealed an elevation in relative mRNA expression of FTH in the Tg and VD+Tg groups compared to the control. Conversely, VD-only treatment and VD post-treatment resulted in lower mRNA expression levels compared to Tg treatment alone. Protein analysis showed a significant decrease in FTH levels after Tg administration and VD post-treatment. Interestingly, the VD-only and VD pre-treated groups exhibited a statistically significant increase in FTH protein expression compared to Tg administration alone. Total iron content analysis was performed to assess the overall iron content within the cells. Although statistically significant differences were not observed, a similar pattern was noted to that observed with FTH levels. Previous research has investigated the interrelationships among iron status, ferritin levels, and VD concentrations [60,61]. The current investigative efforts extend this knowledge base by elucidating the impacts of VD administration on these parameters. The findings contribute to a deeper understanding of the dynamic interactions between micronutrient metabolism and supplementation strategies. They indicate that vitamin D treatment, depending on its timing relative to UPR induction, can modulate the expression proteins involved in cytosolic iron storage (FTH), thereby affecting iron distribution and homeostasis within the organelles, which is a critical parameter in neurodegenerative diseases.

Our research delved into the involvement of cytokines in the regulation of iron metabolism and the UPR, especially in relation to neurodegenerative conditions. Our findings indicate that cytokines such as IL-6 and TNFα can modulate hepcidin expression and influence UPR activation in neurons, suggesting their potential contribution to disease pathogenesis. The measurement of secreted IL-6, IL-8, and TNFα revealed elevated cytokine concentrations in the Tg group, indicating a pro-inflammatory state. Interestingly, both pre- and post-administration of VD resulted in a significant reduction in IL-6 and IL-8 secretion, suggesting a potential anti-inflammatory effect of VD in the context of iron accumulation in neurodegenerative diseases. Pro-inflammatory cytokines significantly increase the amount of hepcidin, which contributes to the retention of iron [62,63]. In contrast, VD reduces the pro-inflammatory cytokines, which in turn regulates iron metabolism. These results indicate that VD may have a beneficial effect on iron metabolism by affecting inflammatory processes.

Investigation into fractalkine, a protein involved in iron metabolism and UPR, demonstrated changes in its gene expression in response to VD treatments. While VD post-treatment resulted in a reduction in fractalkine gene expression compared to Tg-only treatment, VD pre-treatment led to an increase. Additionally, VD alone was found to reduce fractalkine expression and increase its protein secretion, indicating a unique impact of VD on fractalkine regulation. Overall, the results highlight the ability of VD to modulate cytokine secretion and gene expression, along with its influence on fractalkine, suggesting its potential as a therapeutic target for managing neuroinflammation and iron dysregulation in neurodegeneration. Banerjee et al. (2018) found that VD administration resulted in a substantial reduction in the levels of fractalkine, evidenced by a 50% decrease following treatment with TNFα [23]. This observation is in alignment with the data we obtained subsequent to the induction of the UPR, suggesting a consistent modulatory effect of calcitriol on inflammatory mediators.

To our knowledge, the literature provides only a limited number of studies that probe an in-depth analysis of the interactions between VD, the UPR-inducing agent Tg, and iron metabolism. Particularly, studies exploring these relationships beyond the scope of pre-treatment, extending into post-administration phases, are notably absent. Our research aims to elucidate these complex connections, thereby contributing to a more comprehensive understanding of the physiological impact of VD in the context of UPR and iron regulation.

The limitations of our study lie in the fact that they are in vitro experiments using undifferentiated SH-SY5Y cells; this study was conducted with one UPR-inducing agent, and only one type of cell was used to carry out the experiments. The generation and verification of UPR can be best studied with cell cultures. It is our plan to perform further investigations with differentiated cells to see whether they behave in another way for UPR than the undifferentiated ones. As the calcium ion’s role in neurons is not only to provide the proper storage in the ER but also to be involved in cell signaling and neuronal excitation, we will look at other ways of UPR generation and its effect on neuronal behavior. It will be an interesting question whether VD is having the same or different effect in these cases. According to the literature, on some occasions, pre-treatment of VD is more effective in some UPR-producing methods [64]. Iron metabolism is definitely related to UPR as well as VD administration, but we have to be aware that iron movement in the central nervous system involves other cells rather than neurons. Astrocytes and microglia have dominant functions in the regulation of neuronal iron content. We will carry out experiments with neurons co-cultured with glial cells for iron metabolism studies in UPR.

Given the established role of VD in calcium homeostasis, it is imperative that future analyses incorporate an examination of calcium metabolism. This inclusion will enable a more holistic understanding of VD’s biological impact and its potential interactions with other metabolic pathways.

## 4. Materials and Methods

### 4.1. Cell Cultures and Treatments

In our research, SH-SY5Y neuroblastoma cells were cultured in Dulbecco’s Modified Eagle’s Medium/Nutrient Mixture F12 (DMEM/F-12, Lonza Ltd., Basel, Switzerland) with 10% fetal bovine serum (FBS, EuroClone S.p.A., Pero, Italy), 1% nonessential amino acids (NEAA, Lonza Ltd., Basel, Switzerland) and 1% penicillin/streptomycin mixture (P/S 10 k/10 k, Lonza Ltd., Basel, Switzerland) in a humid environment with 5% CO_2_ at 37 °C. For gene expression analysis, 4 × 10^5^ cells were placed on a 6-well plate, while for protein analysis, 10^6^ cells were seeded on a cell culture flask with a 25 cm^2^ surface area. The cells were treated with VD at a final concentration of 25 nM and/or Tg at a final concentration of 25 nM; both were dissolved in absolute ethanol. Four different treatment methods were used during the experiments as follows: VD followed by ethanol, Tg then ethanol, VD then Tg, and Tg before VD. The second agent was administered after 24 h of incubation following treatment with the first agent. The cells were collected after incubation with the second medium for 24 h.

### 4.2. Real-Time qPCR

To analyze gene expression, total RNA was isolated from the collected cells with Aurum Total RNA Mini Kit (Bio-Rad Inc., Hercules, CA, USA) according to the manufacturer’s protocol. The extracted RNA’s concentration was determined with a MultiSkan GO spectrophotometer (Thermo Fisher Scientific Inc., Waltham, MA, USA) using the µDrop plate (Thermo Fisher Scientific Inc., Waltham, MA, USA). Thenceforth, complementary DNA was synthesized from 200 ng of total RNA of each sample with iScript Select cDNA Synthesis Kit (Bio-Rad Inc., Hercules, CA, USA). Gene expression analysis was performed by the real-time quantitative PCR method using CFX96 One Touch Real-Time PCR System (Bio-Rad Inc., Hercules, CA, USA) and iTaq Universal SYBR Green Reagent Mix (Bio-Rad Inc., Hercules, CA, USA). The Livak (∆∆Ct) method was implemented to evaluate data of relative quantification. In every sample, the gene of interest’s expression ratio was determined by normalizing its value against that of β-actin, serving as an internal standard. Throughout the analysis, the expression level of this reference gene was arbitrarily assigned a baseline value of one for comparative purposes. The sequences of primers used in the analysis are listed in Table 1.

### 4.3. Western Blot Analysis

SH-SY5Y cells were collected by centrifugation after treatments. The cells were lysed with 150 µL of ice-cold lysis buffer (50 mM Tris(hydroxymethyl)aminomethane hydrochloride (Tris-HCl), pH 7.4, 150 mM sodium chloride (NaCl), 0.5% Triton-X 100) containing complete mini protease inhibitor cocktail (Roche Ltd., Basel, Switzerland). The protein concentration was determined using a Detergent Compatible Protein Assay Kit (Bio-Rad Laboratories, Hercules, CA, USA), and an equal amount of protein was separated from each sample in 10%, 12%, or 14% polyacrylamide gels. Mini-PROTEAN Tetra Cell polyacrylamide gel electrophoresis system (Bio-Rad Laboratories, Hercules, CA, USA) was used for the vertical electrophoresis. The protein transfer from the gels was carried out by semi-dry electroblotting to nitrocellulose membranes (Pall AG, Basel, Switzerland). Non-fat dry milk (5% (*w*/*v*)) blocking solution (Bio-Rad Laboratories., Hercules, CA, USA) was used for blocking the membranes for 1 h at 25 °C with gentle shaking. Then, the membranes were incubated for 1 h at 25 °C with ferroportin (anti-FPN) IgG (1:1000; Bio-Techne, Minneapolis, MN, USA), the divalent metal transporter-1 (anti-DMT-1) (1:1000, 1 h, room temperature, Thermo Fisher Scientific Inc., Waltham, MA, USA), and the fractalkine (anti-CX3CL1) IgG (1:1000; Thermo Fisher Scientific Inc., Waltham, MA, USA) polyclonal rabbit antibodies. The ferritin heavy chain (anti-FTH) IgG (1:1000; Cell Signaling Technology Europe, Leiden, The Netherlands), the binding immunoglobulin protein (anti-BiP) IgG (1:1000; Cell Signaling Technology Europe, Leiden, The Netherlands), the spliced X-box-binding protein (anti-XBP-1 spliced) (1:1000; Cell Signaling Technology Europe, Leiden, The Netherlands), the endoplasmic reticulum protein 57 (anti-Erp57) IgG (1:1000; Cell Signaling Technology Europe, Leiden, The Netherlands) polyclonal rabbit antibodies, and the C/EBP homologous protein (anti-CHOP) IgG (1:1000; Cell Signaling Technology Europe, Leiden, The Netherlands) polyclonal mouse antibody were overnight incubated at 4 °C. For the loading control, glyceraldehyde 3-phosphate dehydrogenase (GAPDH) (anti-GAPDH IgG, 1:3000; Merck Life Science Kft., Budapest, Hungary) was used in the Western blot protein analysis. Horseradish peroxidase (HRP)-linked goat anti-rabbit IgG was applied (1:3000; Bio-Rad Laboratories, Hercules, CA, USA) as the secondary antibody, and it was incubated for 1 h at 25 °C. The blotted proteins were detected by using WesternBright ECL chemiluminescent substrate (Advansta Inc., San Jose, CA, USA). The membranes were developed using the Alliance Q9 Advanced gel documentation system (UVITEC, Cambridge, UK). The analysis of the protein bands’ optical density was determined by ImageJ 1.53t software [65] and expressed as a target protein/GAPDH ratio. In the presentation of the Western blot data, images have been cropped.

### 4.4. Enzyme-Linked Immunosorbent Assay (ELISA) Measurements

The treatment of SH-SY5Y cells was conducted in accordance with the previously outlined methodology. The cell culture media samples were collected after the treatments and stored at −80 °C until ELISA measurements. TNFα, IL-6, and IL-8 cytokine measurements were determined using human TNFα, IL-6, and IL-8 ELISA assay kits (Invitrogen, Thermo Fisher Scientific Inc., Waltham, MA, USA) according to the manufacturer’s instructions. The absorbance measurements were performed by using a MultiSkan GO microplate spectrophotometer (Thermo Fisher Scientific Inc., Waltham, MA, USA) at 450 nm wavelength. The intensity of the signal is directly proportional to the concentration of the target present in the samples. The concentrations of cytokines were expressed as pg/mL.

### 4.5. Iron Measurements

After the treatments, the cells were harvested by centrifugation and were lysed in 50 mM NaOH for 2 h with gentle shaking at room temperature. An equal volume of 10 mM HCl was added to the samples to neutralize the samples. After neutralization, the samples were mixed with acidic KMnO4 and incubated at 60 °C for 2 h to release protein-bound iron. Reduction of iron and complex formation with ferrozine was carried out in the aqueous solution of 6.5 mM neocuproine, 1 M ascorbic acid, 2.5 M ammonium acetate, and 6.5 mM ferrozine at room temperature for 30 min. The purple ferrozine–iron complexes are detectable at 550 nm. Absorbance readings were obtained through a MultiSkan GO spectrophotometer (Thermo Fisher Scientific Inc., Waltham, MA, USA). Sample concentrations were determined by reference to a FeCl_3_ standard curve and reported as mM iron/mg protein.

### 4.6. Statistical Analysis

Statistical evaluations were performed utilizing SPSS software, version 24.0 (IBM Corporation, Armonk, NY, USA). Once a normal distribution had been confirmed, the significance of the data was determined using a one-way ANOVA with an LSD post hoc test for detailed comparisons. The results are presented as mean ± standard deviation (SD), with a *p*-value below 0.05 denoting statistical significance. In the figures, the asterisks represent a significant difference compared to the control, while the crosses represent a significant difference compared to the Tg treatment.

## 5. Conclusions

In conclusion, this study underscores the intricate relationship between the unfolded protein response, iron metabolism, and VD in neurodegenerative diseases. The research provides insights into the distinct responses of the ERp57/PDIA3 protein, an essential mediator of UPR and a receptor of VD, to various treatment regimens. Notably, VD administration was found to influence UPR activation, iron transporter dynamics, and cytokine secretion, showcasing its capacity to act as a modulator in neuroinflammatory processes and dysregulation of iron homeostasis. The findings illuminate the complex interplay between VD, UPR, and iron metabolism, providing a promising avenue for therapeutic intervention in neurodegenerative conditions characterized by UPR dysregulation and iron accumulation.

## Figures and Tables

**Figure 1 ijms-24-16883-f001:**
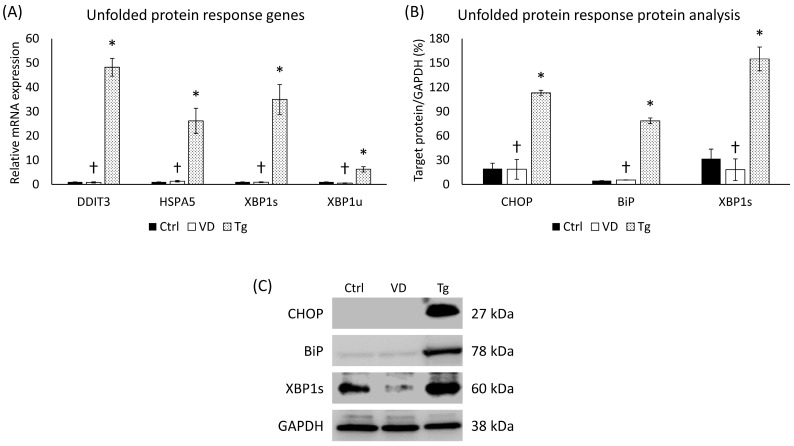
(**A**) Gene expression analysis of UPR-related genes. (**B**) Western blot analysis of UPR-related protein expression. (**C**) Representative Western blots for CHOP, BiP, and XBP1s. Treatments: Ctrl = absolute ethanol for 48 h; Tg = thapsigargin (25 nM) for 24 h, then ethanol for 24 h; VD = vitamin D (25 nM) for 24 h, then ethanol for 24 h. The asterisks represent a significant difference compared to the control, while the crosses represent a significant difference compared to the Tg treatment if the *p*-value was lower than 0.05.

**Figure 2 ijms-24-16883-f002:**
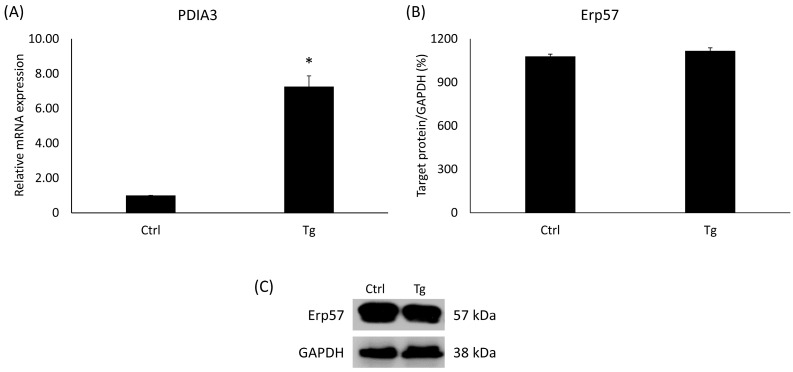
(**A**) Gene expression analysis of PDIA3. (**B**) Western blot analysis of Erp57. (**C**) Representative Western blots for Erp57. Treatments: Ctrl = absolute ethanol for 48 h; Tg = thapsigargin (25 nM) for 24 h, then ethanol for 24 h. The asterisk represents a significant difference compared to the control if the *p*-value was lower than 0.05.

**Figure 3 ijms-24-16883-f003:**
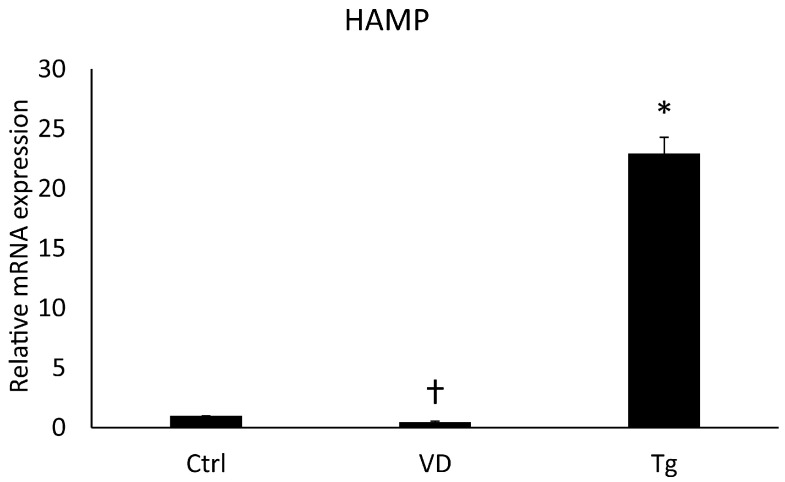
Gene expression analysis of HAMP gene. Treatments: Ctrl = absolute ethanol for 48 h; Tg = thapsigargin (25 nM) for 24 h, then ethanol for 24 h; VD = vitamin D (25 nM) for 24 h, then ethanol for 24 h. The asterisk represents a significant difference compared to the control, while the cross represents a significant difference compared to the Tg treatment if the *p*-value was lower than 0.05.

**Figure 4 ijms-24-16883-f004:**
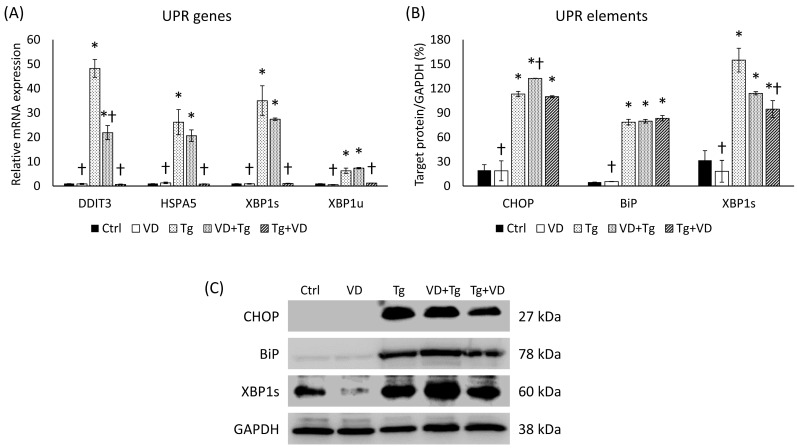
(**A**) Gene expression analysis of UPR-related genes in co-treatments. (**B**) Western blot analysis of UPR elements in co-treatments. (**C**) Representative Western blots for CHOP, BiP, and XBP1s. Treatments: Ctrl = absolute ethanol for 48 h; Tg = thapsigargin (25 nM) for 24 h, then ethanol for 24 h; VD = vitamin D (25 nM) for 24 h, then ethanol for 24 h; VD+Tg = vitamin D (25 nM) for 24 h, then thapsigargin (25 nM) for 24 h; Tg+VD = thapsigargin (25 nM) for 24 h, then vitamin D (25 nM) for 24 h. The asterisks represent a significant difference compared to the control, while the crosses represent a significant difference compared to the Tg treatment if the *p*-value was lower than 0.05.

**Figure 5 ijms-24-16883-f005:**
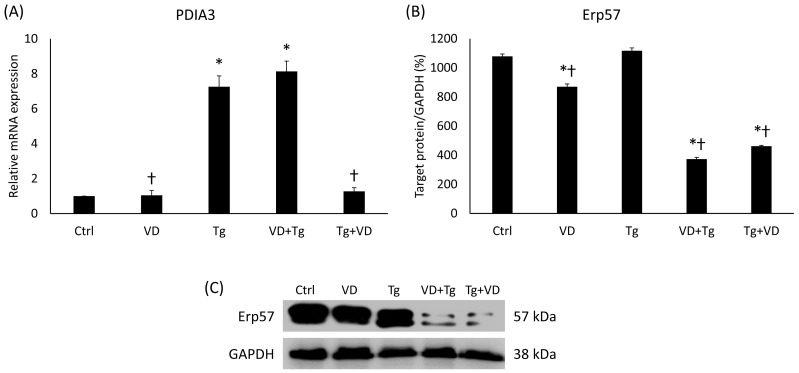
(**A**) Gene expression analysis of PDIA3 gene in co-treatments. (**B**) Western blot analysis of Erp57 in co-treatments. (**C**) Representative Western blots for Erp57. Treatments: Ctrl = absolute ethanol for 48 h; Tg = thapsigargin for 24 h, then ethanol for 24 h; VD = vitamin D for 24 h, then ethanol for 24 h; VD+Tg = vitamin D for 24 h, then thapsigargin for 24 h; Tg+VD = thapsigargin for 24 h, then vitamin D for 24 h. The asterisks represent a significant difference compared to the control, while the crosses represent a significant difference compared to the Tg treatment if the *p*-value was lower than 0.05.

**Figure 6 ijms-24-16883-f006:**
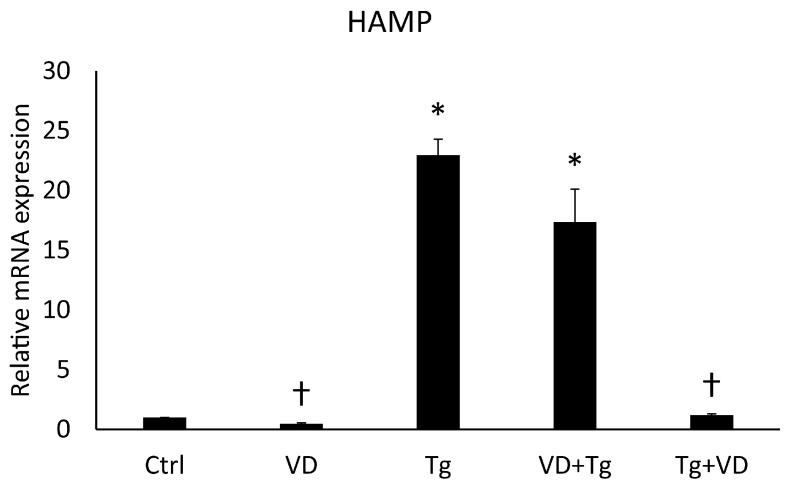
Gene expression analysis of HAMP. Treatments: Ctrl = absolute ethanol for 48 h; Tg = thapsigargin for 24 h, then ethanol for 24 h; VD = vitamin D for 24 h, then ethanol for 24 h; VD+Tg = vitamin D for 24 h, then thapsigargin for 24 h; Tg+VD = thapsigargin for 24 h, then vitamin D for 24 h. The asterisks represent a significant difference compared to the control, while the crosses represent a significant difference compared to the Tg treatment if the *p*-value was lower than 0.05.

**Figure 7 ijms-24-16883-f007:**
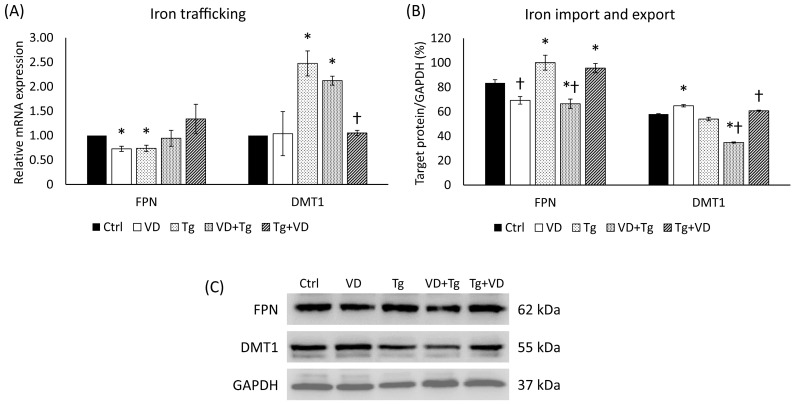
(**A**) Gene expression analysis of FPN and DMT-1 gene in co-treatments. (**B**) Western blot analysis of FPN and DMT-1 in co-treatments. (**C**) Representative Western blots for FPN and DMT1. Treatments: Ctrl = absolute ethanol for 48 h; Tg = thapsigargin for 24 h, then ethanol for 24 h; VD = vitamin D for 24 h, then ethanol for 24 h; VD+Tg = vitamin D for 24 h, then thapsigargin for 24 h; Tg+VD = thapsigargin for 24 h, then vitamin D for 24 h. The asterisks represent a significant difference compared to the control, while the crosses represent a significant difference compared to the Tg treatment if the *p*-value was lower than 0.05.

**Figure 8 ijms-24-16883-f008:**
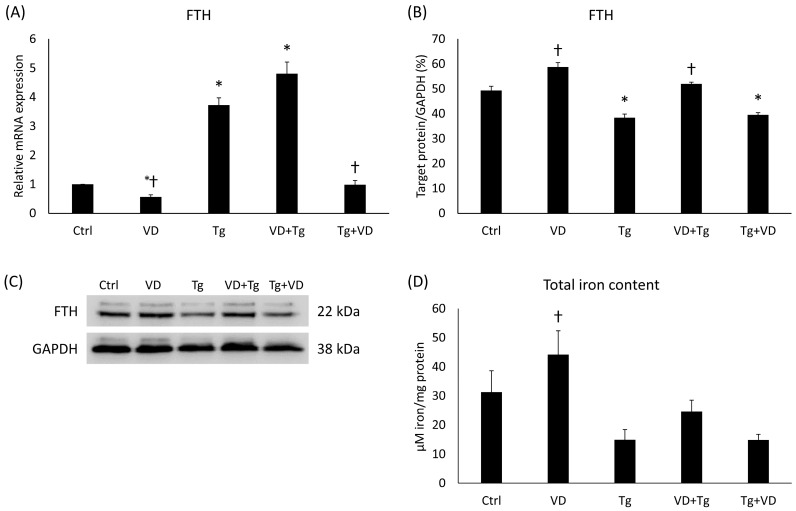
(**A**) Gene expression analysis of FTH gene in co-treatments. (**B**) Western blot analysis of FTH in co-treatments. (**C**) Representative Western blots for FTH. (**D**) Measurement of the total iron content in co-treatments. Treatments: Ctrl = absolute ethanol for 48 h; Tg = thapsigargin for 24 h, then ethanol for 24 h; VD = vitamin D for 24 h, then ethanol for 24 h; VD+Tg = vitamin D for 24 h, then thapsigargin for 24 h; Tg+VD = thapsigargin for 24 h, then vitamin D for 24 h. The asterisks represent a significant difference compared to the control, while the crosses represent a significant difference compared to the Tg treatment if the *p*-value was lower than 0.05.

**Figure 9 ijms-24-16883-f009:**
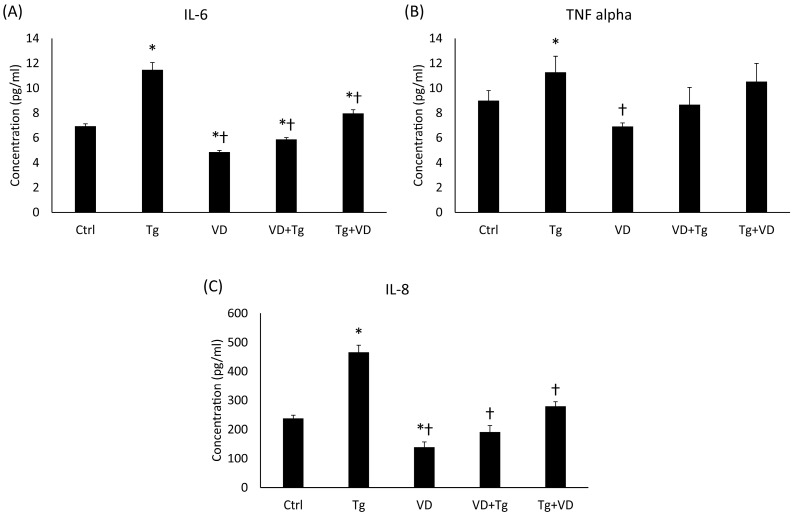
The ELISA measurements of pro-inflammatory cytokines. (**A**) Concentration of IL-6. (**B**) Concentration of TNFα. (**C**) Concentration of IL-8. Treatments: Ctrl = absolute ethanol for 48 h; Tg = thapsigargin for 24 h, then ethanol for 24 h; VD = vitamin D for 24 h, then ethanol for 24 h; VD+Tg = vitamin D for 24 h, then thapsigargin for 24 h; Tg+VD = thapsigargin for 24 h, then vitamin D for 24 h. The asterisks represent a significant difference compared to the control, while the crosses represent a significant difference compared to the Tg treatment if the *p*-value was lower than 0.05.

**Figure 10 ijms-24-16883-f010:**
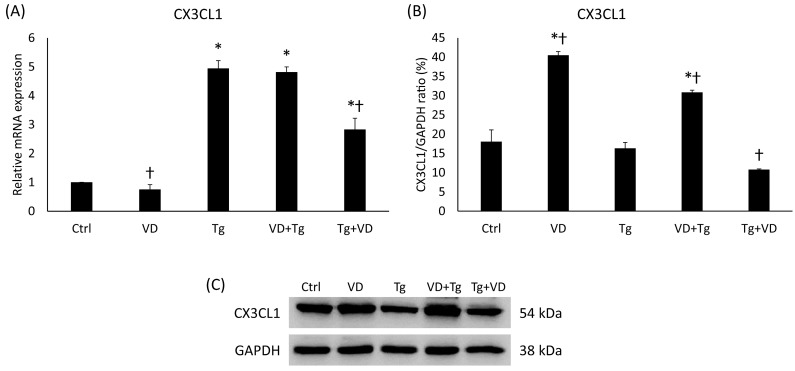
(**A**) Gene expression analysis of CX3CL1 gene in co-treatments. (**B**) Western blot analysis of CX3CL1 in co-treatments. (**C**) Representative Western blots for CX3CL1. Treatments: Ctrl = absolute ethanol for 48 h; Tg = thapsigargin for 24 h, then ethanol for 24 h; VD = vitamin D for 24 h, then ethanol for 24 h; VD+Tg = vitamin D for 24 h, then thapsigargin for 24 h; Tg+VD = thapsigargin for 24 h, then vitamin D for 24 h. The asterisks represent a significant difference compared to the control, while the crosses represent a significant difference compared to the Tg treatment if the *p*-value was lower than 0.05.

**Figure 11 ijms-24-16883-f011:**
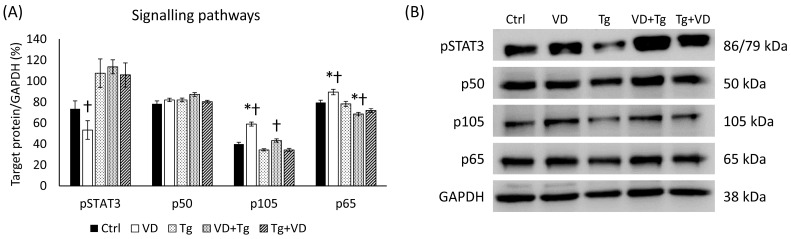
(**A**) Western blot analysis of phosphorylated signaling pathway elements in co-treatments. (**B**) Representative Western blots for pSTAT3, p50, p105 and p60. Treatments: Ctrl = absolute ethanol for 48 h; Tg = thapsigargin for 24 h, then ethanol for 24 h; VD = vitamin D for 24 h, then ethanol for 24 h; VD+Tg = vitamin D for 24 h, then thapsigargin for 24 h; Tg+VD = thapsigargin for 24 h, then vitamin D for 24 h. The asterisks represent a significant difference compared to the control, while the crosses represent a significant difference compared to the Tg treatment if the *p*-value was lower than 0.05.

**Figure 12 ijms-24-16883-f012:**
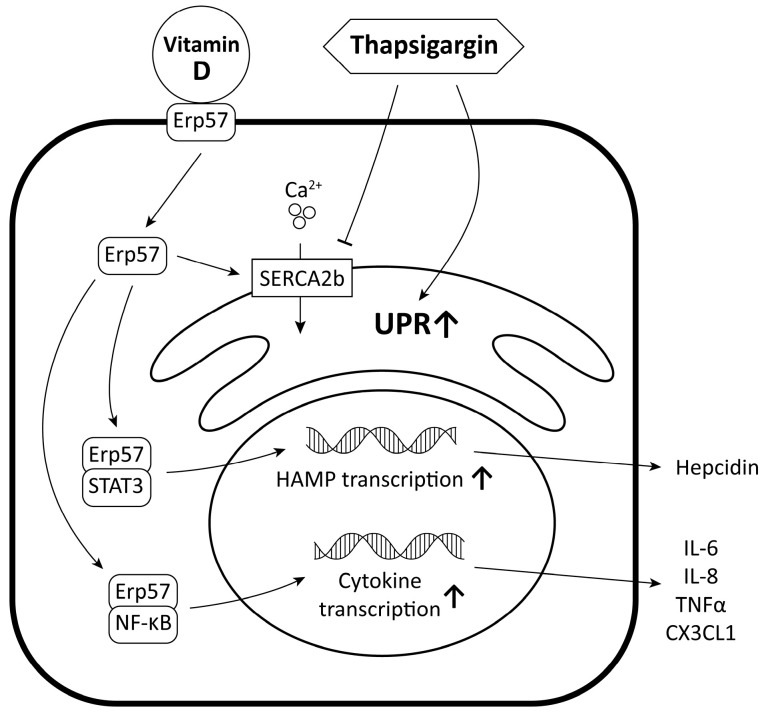
Molecular relationship between UPR induced by thapsigargin, vitamin D, and iron homeostasis.

**Table 1 ijms-24-16883-t001:** Real-time qPCR primer sequence list.

Primer	Sequence 5′→3′
β-actin forward	CCCGCGAGTACAACCTTCTT
β-actin reverse	TCATCCATGGCGAACTGGTG
CX3CL1 forward	TACCTGTAGCTTTGCTCATC
CX3CL1 reverse	GTCTCGTCTCCAAGATGATT
DDIT3 forward	AATGAAAGGAAAGTGGCACA
DDIT3 reverse	ATTCACCATTCGGTCAATCA
DMT-1 forward	GTGGTTACTGGGCTGCATCT
DMT-1 reverse	CCCACAGAGGAATTCTTCCT
FPN forward	TTCCTTCTCTACCTTGGTCA
FPN reverse	AAAGGAGGCTGTTTCCATAG
FTH forward	GAGGTGCCCGAATCTTCCTTC
FTH reverse	TCAGTGGCCAGTTTGTGCAG
HAMP forward	CAGCTGGATGCCCATGTT
HAMP reverse	TGCAGCACATCCCACACT
HSPA5 forward	GTCCCACAGATTGAAGTCAC
HSPA5 reverse	CGATTTCTTCAGGTGTCAGG
PDIA3 forward	TGTGGTCACTGTAAGAACCT
PDIA3 reverse	ATCCATCTTGGCTATGACGA
XBP-1 unspliced forward	TGAGAACCAGGAGTTAAGACA
XBP-1 unspliced reverse	AGAGGTGCACGTAGTCTG
XBP-1 spliced forward	GCTTAGTCCGCAGCAGGT
XBP-1 spliced reverse	GAGTCAATACCGCCAGAATC

## Data Availability

The data underlying this article are available in the article.

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
