# Peer review of "Interplay of Vitamin D, Unfolded Protein Response, and Iron Metabolism in Neuroblastoma Cells: A Therapeutic Approach in Neurodegenerative Conditions"

_ijms, 2023, doi:10.3390/ijms242316883_

Round 1

Reviewer 1 Report

Comments and Suggestions for Authors

Reviewer comments and suggestions

The authors in this study used SH-SY5Y neuroblastoma cells to investigate the relationship between Unfolded Protein Response (UPR), iron metabolism, and Vitamin D3 (VD). The study results indicated that VD affects UPR pathways, cytokine release, and iron-related genes, potentially offering anti-inflammatory benefits. Furthermore, pro-inflammatory cytokines like interleukin-6 (IL-6) and tumor necrosis factor-alpha (TNFα) were impacting UPR activation in cells. The study result clues the intricate connections among VD, UPR, and iron metabolism in SH-SY5Y neuroblastoma cells.

Overall, the manuscript is well written. I have listed the concerns and comments that needed to be explained or modified.

  1. Lines 67-68 The authors could explain it more about how
  2. Line 82-83 These references should be discussed for the common reader
  3. Comments for figure 11 p105 (in VD) group the band intensity was higher but the histogram shows a lower value than the control. How possible Please check again
  4. Also please confirm all with these kinds of mistakes in other histograms and Western results.
  5.  First paragraph of discussion section: “The first paragraph require the novelty of this study rather than discussing other findings”
  6. Line 298-300 The discussion may consist of a ray diagram for the signaling pathway they observed in this study, it could provide the mechanism
  7. Line 311-313 For therapeutic purposes, the authors did not make any contribution; only writing in the manuscript does not carry any value, better they can explain with examples or some other fitted studies.
  8. Line 315-316 I think there would be some animal studies that focus on this.
  9. Line 331-333 Please mention the table or figure in the text for easy understanding. Similar comments for lines 356-358

Reviewer 2 Report

Comments and Suggestions for Authors

I feel that this paper has been carefully drafted and the research has been carried out appropriately. There is nothing to point out regarding the research data. However, please improve the inappropriately small font size in the diagram.

Also, if possible, I think it would be better to discuss data on the intracellular distribution of iron or possible relationships with intracellular organelles.
